# Co-Cultivation of *Fusarium*, *Alternaria*, and *Pseudomonas* on Wheat-Ears Affects Microbial Growth and Mycotoxin Production

**DOI:** 10.3390/microorganisms9020443

**Published:** 2021-02-20

**Authors:** Annika Hoffmann, Gunnar Lischeid, Matthias Koch, Peter Lentzsch, Thomas Sommerfeld, Marina E. H. Müller

**Affiliations:** 1Leibniz Centre for Agricultural Landscape Research (ZALF), 15374 Müncheberg, Germany; lischeid@zalf.de (G.L.); lentzsch@zalf.de (P.L.); mmueller@zalf.de (M.E.H.M.); 2Institute for Horticultural Sciences, Humboldt-Universität zu Berlin, 14195 Berlin, Germany; 3Berlin-Brandenburg Institute of Advanced Biodiversity Research (BBIB), 14195 Berlin, Germany; 4Institute for Environmental Sciences and Geography, University of Potsdam, 14476 Potsdam, Germany; 5Bundesanstalt für Materialforschung und -prüfung (BAM), 12205 Berlin, Germany; matthias.koch@bam.de (M.K.); thomas.sommerfeld@bam.de (T.S.)

**Keywords:** *Alternaria*, antagonists, *Fusarium*, microbe interactions, mycotoxins, priority effect, *Pseudomonas*, SOM-SM, wheat

## Abstract

Mycotoxigenic fungal pathogens *Fusarium* and *Alternaria* are a leading cause of loss in cereal production. On wheat-ears, they are confronted by bacterial antagonists such as pseudomonads. Studies on these groups’ interactions often neglect the infection process’s temporal aspects and the associated priority effects. In the present study, the focus was on how the first colonizer affects the subsequent ones. In a climate chamber experiment, wheat-ears were successively inoculated with two different strains (*Alternaria tenuissima At*625, *Fusarium graminearum Fg*23, or *Pseudomonas simiae Ps*9). Over three weeks, microbial abundances and mycotoxin concentrations were analyzed and visualized via Self Organizing Maps with Sammon Mapping (SOM-SM). All three strains revealed different characteristics and strategies to deal with co-inoculation: *Fg*23, as the first colonizer, suppressed the establishment of *At*625 and *Ps*9. Nevertheless, primary inoculation of *At*625 reduced all of the *Fusarium* toxins and stopped *Ps*9 from establishing. *Ps*9 showed priority effects in delaying and blocking the production of the fungal mycotoxins. The SOM-SM analysis visualized the competitive strengths: *Fg*23 ranked first, *At*625 second, *Ps*9 third. Our findings of species-specific priority effects in a natural environment and the role of the mycotoxins involved are relevant for developing biocontrol strategies.

## 1. Introduction

Without crop protection, the yield loss of one of the world’s most important cereals, wheat (*Triticum aestivum* L.), is approximately 50% [1]. However, even with plant protection (mechanical, biological, chemical), the potential loss reduces to only about 29%. Approximately one-third of this can be explained by plant diseases [1,2]. One of the major wheat cultivation diseases is Fusarium head blight (FHB). It is caused by a complex of about 19 *Fusarium* (F.) species, of which *F. graminearum* [Schwabe (teleomorph *Gibberella zeae* (Schwein) Petch)] and *F. culmorum* [W.G. Smith Sacc (teleomorph unknown)] are the most virulent ones [3,4]. The symptoms of FHB include wrinkling and reduced size and weight of the kernels. These symptoms lead to reduced yields, modified grain quality, and reduced seed germination [5,6].

Further important fungal wheat pathogens belong to the genus *Alternaria* (A.), which ubiquitously occur on wheat-ears and leaves. It is commonly associated with diseases such as black point, black kernel, and leaf blight, mainly caused by the two species *A. alternata* and *A. tenusissima* [7,8]. It can affect crops on the field or plant products at the harvest or post-harvest stage [7,9,10].

Fungi of both genera are capable of reducing yield and grain quality and producing mycotoxins. In the *Fusarium* genus, especially *F. graminearum* and *F. culmorum* produce toxicologically relevant classes of mycotoxins: deoxynivalenol (DON), nivalenol (NIV), and zearalenone (ZEN) [11,12]. In humans and animals, they can damage the digestive tract, the liver, the endocrine system, the blood circulatory system, and alters the immune response. They can cause several acute and chronic diseases and make wheat products unsuitable for consumption if their concentration exceeds the maximum levels set by the European Commission [11,13,14,15,16,17].

Compared to the *Fusarium* mycotoxins, *Alternaria* mycotoxins have received less attention. Nevertheless, several recent studies on the toxicological effects of these mycotoxins brought this group of toxins into the focus of consumer protection and agricultural science [18,19,20]. The most studied and toxicologically relevant mycotoxins produced by *Alternaria* are alternariol (AOH), alternariol monomethyl ether (AME), altenuene (ALT), the perylene quinone derivatives altertoxin (ATX) I and II, and tenuazonic acid (TeA) [7,21,22]. All these mycotoxins are, such as the *Fusarium* mycotoxins, harmful to humans and animals. They have been reported to be related to oesophageal cancer and have genotoxic, cytotoxic, and estrogenic potential as well as to induce oxidative stress, autophagy, and senescence in vitro [19,23,24,25,26,27].

Interactions between co-occurring genera such as *Fusarium* and *Alternaria* are crucial for the fungal community formation and co-existence [28,29,30,31]. Most of these relationships are generally considered competitive and are associated with the occurrence of mycotoxins [31]. Experiments with coinocu- lation showed that weaker fungi develop poorly in the presence of strong competitors, which can lead to the exclusion of one of the competing species or even to the complete absence of co-existence [32,33,34].

*Fusarium* and *Alternaria* interactions are described as complex, depending on the type of mycotoxins involved in the competition [33,35,36]. Usually, a considerably increased mycotoxin production is shown with simultaneous infection with *Alternaria* and *Fusarium* [32,37,38]. Xu et al. [31] concluded that the observed increase in mycotoxins under mixed inoculations could be due to the fungi’ stress during competition. Gannibal [35] studied the competition of *Alternaria* with other fungal genera, including fusaria, and observed that the species *F. langsethiae*, *F. tricinctum*, and *F. graminearum* grew faster while suppressing the growth of *Alternaria*. Another study by Xu et al. [39] also concluded that fusaria usually reduce the other fungi’ abundance in wheat-ears, which leads to a strong change in the fungal community structure.

Competitors to the genera *Fusarium* and *Alternaria* are, among others, bacteria of the genus *Pseudomonas* (P.) [40,41]. Many strains of *Pseudomonas* produce inhibitory or lethal metabolites and, thus are capable of antibiosis. Secondary metabolites, such as phloroglucinols, phenazines, pyoluteorin, pyrrolnitrin, and hydrogen cyanide produced by pseudomonads are described to suppress soil-borne pathogens [42]. Pseudomonads are also found on wheat-ears as antagonists against phytopathogenic fungi [43,44]. In a field trial, wheat and barley were inoculated with *F. culmorum* and then treated with either one of two different *P. fluorescence* strains [45]. Both treatments resulted in a significant reduction of up to 78% of DON levels as well as a diminished disease-associated loss in 1000-grain weight in both cereals [45]. Spray inoculation of wheat-ears with an antagonistic strain of *P. simiae* resulted in reduced contents of DON and ZEN as well as ALT and TeA produced by natural occurring fusaria and alternaria in experimental field plots [46].

Important aspects of the infection process, which so far were neglected in most studies, are the infection’s temporal succession, immigration processes, and priority effects. Community meta- barcoding has shown how fungal community composition changes during kernel development. Basidiomycete yeasts dominate endophyte communities before anthesis. However, during kernel development, they are replaced by a more opportunistic ascomycete-rich community [47]. *Fusarium* spp. interrupt these dynamics by excluding other fungi from floral tissues, leading to reduced community diversity, especially in the kernels [47,48]. Endophytes such as *Cladosporium*, *Itersonillia*, and *Holtermaniella* correlated negatively with *Fusarium* species by out-competing or preventing the spread of FHB [47].

However, metabarcoding data can only reveal the dynamics of community composition but cannot provide information about the underlying mechanisms during the infection phase. To investigate this topic, more detailed studies at the species level are needed. For this reason, our research focuses on three microbial species that occur simultaneously on wheat-ears. Three strains from the genera *Alternaria*, *Fusarium*, and *Pseudomonas* were selected for experimental implementation. Fusaria usually infects the wheat-ear around ear emergence and the start of anthesis [49]. *Alternaria* belongs to the saprotrophs or opportunistic pathogens that cause black spot on wheat during ripening ears [50,51]. Therefore, its appearance is more dependent on the state of the plant and the weather conditions. Plant-associated pseudomonads usually derive from the rhizosphere, the soil compartment adjacent to and surrounding the root [52,53]. *Pseudomonas* can also enter and leave the above-ground plant parts by spreading through the air, rainwater, and insect vectors [54,55,56].

This work is dedicated to the influences that the first colonizer has on the subsequent one. Is a well established primary population capable of reducing the establishment of a subsequent population? Is it also capable of inhibiting the production of secondary metabolites such as mycotoxins?

To clarify the effect of temporally phased colonization on coexisting microorganisms on wheat-ears, an inoculation experiment was set up. Over three weeks, the growth and mycotoxin production of two successively inoculated strains of the genera *Pseudomonas*, *Alternaria*, or *Fusarium* were monitored. A novel approach in the use of Self-Organizing Maps (SOM) with Sammon Mapping (SM) was then used to visualize the dependencies in the establishment of the strains.

Overall, we aim to understand the interactions of pathogens and antagonists better on wheat, its immigration processes, priority effects, and the role of the mycotoxins involved. This knowledge may ultimately help develop effective strategies to control fungal pathogens.

## 2. Materials and Methods

### 2.1. Fungal and Bacterial Isolates

Isolates used in this experiment originated from wheat plants of commercial farms located in the state of Brandenburg (Germany): *A. tenuissima At*625 (*At*625), *F. graminearum Fg*23 (*Fg*23) and *P. simiae Ps*9 ${}^{rif+/kan+}$ (*Ps*9) [32,33,57]. Both fungi were selected because of their proven high pathogenic activity and mycotoxin production in laboratory tests [12,32]. The *Ps*9 hosts the gene prnCto that synthesize the antibiotic pyrrolnitrin and showed in in vitro tests antagonistic potential against alternaria and fusaria, with the ability to reduce the mycotoxin production [40,46]. Fungi are stored at −20 ${}^{\circ }$C as single-spore cultures on sterile wheat kernels. *Ps*9 is stored in a Cryobank tube (Mast Diagnostica, Reinfeld/Germany) at − 20 ${}^{\circ }$C. All three isolates are registered in the culture collection of microorganisms at the Leibniz-Centre for Agricultural Landscape Research Müncheberg, Germany.

### 2.2. Inoculum Production

Fungal inocula were produced by placing individual kernels with mycelium and conidia from stock culture in Petri dishes (*ϕ* 9 cm) onto Synthetic Nutrient Agar (SNA; [58]) for *Fusarium* and Potato Carrot Agar (PCA; [59]) for *Alternaria*. The plates were preincubated for 4 days at 25 ${}^{\circ }$C in the dark. Afterward, an incubation at room temperature under mixed black light (near UV, emission 310–360 nm) and artificial daylight with a photoperiod 12 h light: 12 h dark for 10 days was followed. Conidial suspension for inoculation of wheat-ears was obtained by washing culture surface with sterile 1/4 strength Ringer’s solution and filtering the resulting suspension through two-layered mull. The conidial concentration was determined using a Thoma counting chamber of 0.1 mm depth (Poly-Optik GmbH, Bad Blankenburg, Germany) and adjusted to a density of $2\times 10^{5}$ conidia mL${}^{-1}$.

The inoculum preparation of *P. simiae* 9 is described in detail by Müller et al. [40]. The suspension density was determined using a Thoma counting chamber of 0.01 mm depth (Poly-Optik GmbH, Bad Blankenburg, Germany) and adjusted to $5\times 10^{6}$ cells mL${}^{-1}$ by using sterile 1/4 strength Ringer’s solution.

### 2.3. Experimental Design

The experiment was conducted under climate chamber conditions. Summer wheat cultivar Tybalt, which is highly susceptible to FHB, was used for the experiment. Seeds were sown in pots (11 cm × 11 cm × 12 cm) containing autoclaved quartz sand (*ϕ* 0.71–1.25 mm). Pots were placed in a growth chamber (KTLK 2000; Nema, Netzschkau/Vötsch, Germany) at a day/night period described in Table 1.

Plants were fertilized with 10 mL per pot with Knop’s nutrient solution [60] every week. A total of 250 wheat-ears were available for the experiment. The experiment was set up with nine to ten replications (wheat-ears). Pots with the same first inoculant stayed in separate identical climate chambers (Fitotron^®^, Weiss Technik, Loughborough, UK), running the same program (Table 1). The second and third inoculation were carried out randomly. In Figure 1, the experimental procedure can be followed, and in Figure 2 the success of the infected wheat-ears can be seen.

The inoculum was applied by using a spray bottle. Each ear was treated with 1 mL suspension of either *At*625, *Fg*23, or *Ps*9. Controls were treated with 1 mL of sterile 1/4 strength Ringer’s solution instead. After the inoculation each wheat-ear was covered with a polyethylene bag.

One week after the first inoculation, each ear was treated with another strain. For example, if *Alternaria* was applied to the ear in the primary inoculation, *Fusarium* or *Pseudomonas* was now applied. Again 1 mL of the corresponding suspension was sprayed on the ears. Sterile 1/4 strength Ringer’s solution served as the control treatment. The ears were sealed with a new polyethylene bag after this inoculation.

Two weeks after the first inoculation and correspondingly one week after the previous inoculation, each ear was treated with the same isolate used for the second inoculation. Afterwards, they were covered in new plastic bags. For example, if *Fusarium* was applied in the previous (second) inoculation, *Fusarium* was applied again to give the presumably disadvantaged opponent an advantage. The procedure was the same as in the previous inoculations.

Over the experiment time, samples were also taken from wheat plants that were only treated with 1/4 strength Ringer’s solution and stayed separated in the origin climate chamber. Therefore a control group, for possibly already present *Fusarium*, *Alternaria*, or *Pseudomonas* brought with the seed, was provided (sample ID only consists of C’s). Besides, a control group that was only treated in the second and third inoculation phase was available and therefore had no first colonizer in the strict sense (sample ID begins with C).

After three weeks, the experiment was closed. Over the experiment’s period, samples were repeatedly taken to compare growing colonies with the isolate applied. Therefore, suspensions obtained from the wheat-ears were spread out on SNA or PCA plates for fungal determination and on King’s B agar with 0.4 g L${}^{-1}$ cycloheximide supplemented by rifampin and kanamycin (100 g mL${}^{-1}$ and 50 g mL${}^{-1}$, respectively) to record *P. simiae* 9 ${}^{rif+/kan+}$ exclusively.

At each point in time: one week after the first inoculation (with one isolate), two weeks after the first inoculation (with two isolates), and three weeks after the first inoculation (with two isolates; the second one applied twice), wheat-ears were cut off.

After harvesting, individual wheat-ears were dried at 60 ${}^{\circ }$C for three days and subsequently ground in the mixer mill (Mixer Mill MM 200, Retsch GmbH, Haan, Germany) for 2 min at 20 s${}^{-1}$. Three to four milled wheat-ears of the same variant were then combined and homogenized. This step was executed to reach about the same biomass for every variant and brought small and larger wheat-ears together. In the end, three to four pooled samples per variant were available for further qPCR and HPLC-MS/MS analysis.

### 2.4. Quantitative Analyses Via qPCR

50 mg of dried and grounded material were used for genomic DNA extraction by using the DNeasy Plant Mini kit (QIAGEN GmbH, Hilden, Germany). The extraction is described in detail by Müller et al. [40]. The quantification of fungal genome copies of *Fusarium* and *Alternaria* by qPCR was also described in detail by Müller et al. [40]. The quantification of *Pseudomonas* was based on the primers and probes described by Bergmann et al. [61]. The PCR conditions were adapted to a two-step PCR: 10 min on 95 ${}^{\circ }$C followed by 45 cycles of 95 ${}^{\circ }$C for 30 s and 62 ${}^{\circ }$C for 60 s. The reactions contained 4 L GC probe master mix (Solis biodyne, 50411 Tartu, Estonia), 770 nM forward and reverse primer, 100 nM probe, and 1 L sample DNA. The detection of *Pseudomonas* was based on the region between the primers Pse435F and Pse686R of the target region V3–V4 in the 16S rRNA gene sequence (Table 2).

All qPCR assays contained negative controls, and all measurements were done in duplicate. Tenfold serial dilutions of extracted genomic DNA from pure cultures of *Ps*9 served as verification. Different strains of plant-associated bacteria species were used as negative controls: Strains of *Stenotrophomonas rhizophila* S1-C57-R, *Xanthomonas campestris* DSM 1050, *Acinetobacter calcoaceticus* H1-C10-RR, *Sphingomonas paucimobilis* DSM 1098, and *Pseudoxanthomonas indica* H2–E14, which were conserved in the culture collection of microorganisms at the Leibniz Center for Agricultural Landscape Research Müncheberg, Germany.

### 2.5. Mycotoxin Analyses Via HPLC-MS/MS

#### 2.5.1. Extraction of Wheat Samples

0.5–2 g of milled wheat samples were weighed into 15 mL centrifugation tubes and extracted with 12 mL methanol/water (60/40 *v/v*) by ultrasonication for 30 min (DT 255, Bandelin electronic GmbH & Co KG, Berlin, Germany) followed by horizontal shaking for 30 min (IKA HS 501 horizontal shaker, IKA, Staufen, Germany). After centrifugation for 10 min (3400 rpm corresponds to 5000 g; Sigma 6K15 centrifuge; Sigma Zentrifugen, Osterode am Harz, Germany), 1 mL of the clear extract was transferred into each of two HPLC vials. To one HPLC vial, 100 L internal standard solution for *Fusarium* toxins (IS-F) was added containing ${}^{13}C_{15}$-DON, ${}^{13}C_{17}$-3-Ac-DON, ${}^{13}C_{15}$-NIV and ${}^{13}C_{18}$-ZEN; to the other HPLC vial, 100 L internal standard solution for *Alternaria* toxins (IS-A) was added containing D3-ALT, D3-AOH, and D3-AME.

#### 2.5.2. Analytical Standards and Calibration

All solvents were provided in analytical grade. Deionized water was supplied by a Purelab flex 2 (ELGA LabWater, Celle, Germany). Pure analytical standards (in acetonitrile) of DON, 15-Ac-DON, 3-Ac-DON, NIV, ZEN, TeA, AOH, AME and Tentoxin (TEN) (100 g mL${}^{-1}$), DON-3G (50 g mL${}^{-1}$) and ${}^{13}C_{15}$-DON, ${}^{13}C_{15}$-NIV, ${}^{13}C_{17}$-3-Ac-DON and ${}^{13}C_{18}$-ZEN (25 g mL${}^{-1}$) were obtained from Romer Labs (Tulln, Austria). Solid standards of ALT, D3-ALT, D3-AOH, and D3-AME were purchased from HPC Standard GmbH (Cunnersdorf, Germany). From these solid standards, single analyte solutions were prepared for each native and isotopic labeled standard by dilution with methanol/water (60/40 *v/v*). Multicomponent solutions for constructing calibration curves of *Fusarium* toxins and *Alternaria* toxins were prepared by mixing of the single analyte standards and dilution with methanol/water (60/40 *v/v*); six calibration levels were prepared in the range of 10 to 1000 ng mL${}^{-1}$ for each analyte. All solutions were stored at −20 ${}^{\circ }$C in the dark.

#### 2.5.3. HPLC-MS/MS Conditions

HPLC–MS/MS analyses were done using an API 4000 QTrap^®^ MS/MS system (AB Sciex, Darmstadt, Germany), equipped with an ESI interface and hyphenated to a 1200 series HPLC system comprising a degasser, a binary pump, autosampler, and a column oven from Agilent Technologies (Waldbronn, Germany). The chromatographic separation of 20 L injected sample was achieved using an Eurospher II 100-5 C18 P analytical column (250 mm × 4 mm, 5 m particles; Knauer, Berlin, Germany), preceded by an Eurospher II 100-5 C18 P guard column (4 mm × 2 mm, 5 m particles). The column oven was set to 30 ${}^{\circ }$C and the flow rate of the mobile phase was 500 L min${}^{-1}$. The mobile phase was a time-programmed gradient with the following conditions for *Fusarium* toxins: A (water, 0.1% acetic acid) and B (methanol, 0.1% acetic acid). The mobile phase gradient consisted of 0–10 min 15% B, 10–19 min ramp to 100% B, 19–25 min hold at 100% B, 25–28 min ramp back to initial conditions (equilibration time: 7 min). For *Alternaria* toxins: A (water, 5 mmol L${}^{-1}$ NH${}_{4}$Ac + NH${}_{4}$OH to pH = 8.7) and B (methanol, 5 mmol L${}^{-1}$ NH${}_{4}$Ac). The mobile phase gradient consisted of 0–5 min 10% B, 5–15 min ramp to 100% B, 15–22 min hold at 100% B, 22–22.5 min ramp back to initial conditions (equilibration time: 7.5 min). The column effluent was directly transferred into the ESI interface without splitting. The ESI interface was operated in negative ion mode, and analyses of the mycotoxins were carried out using the multiple reaction monitoring mode. The optimized conditions for each mycotoxin are summarized in Table 3.

The ANALYST 1.6.2 software package (AB Sciex Pte. Ltd., Darmstadt, Germany) was used to control the HPLC–MS/MS system as well as for data acquisition and processing of quantitative data obtained from standard calibration and samples.

#### 2.5.4. Quantification, Performance Limits and Quality Control

After linear regression of the external 6-point calibration lines constructed for each analyte, quantification of the mycotoxin contents in the wheat samples was done using the corresponding isotopic labeled IS. When not available, the following evaluation procedures were used: 15-Ac-DON via ${}^{13}C_{17}$-3-Ac-DON, DON-3G via ${}^{13}C_{15}$-DON, and TeA via D3-ALT. Limits of detection (LOD) and limits of quantification (LOQ) were calculated for each analyte based on the linear regression line model according to DIN 32645. LOD/LOQ values were found to be 3/12 g kg${}^{-1}$ for DON, 11/42 g kg${}^{-1}$ for 3-Ac-DON, 6/24 g kg${}^{-1}$ for 15-Ac-DON, 2/8 g kg${}^{-1}$ for DON-3G, 5/19 g kg${}^{-1}$ for NIV, 7/28 g kg${}^{-1}$ for ZEN, 2/8 g kg${}^{-1}$ for ALT, 2/9 g kg${}^{-1}$ for AOH, 0.3/1.2 g kg${}^{-1}$ for AME and 12/42 g kg${}^{-1}$ for TeA. For quality control of the *Fusarium* mycotoxins, the certified reference material ERM^®^-BC600 (wheat flour) was used with the certified values for DON: 102±11 g kg${}^{-1}$, NIV: 1000±130 g kg${}^{-1}$ and ZEN: 90±8 g kg${}^{-1}$.

### 2.6. Data Management

Research data is available at the ZALF data storage: https://www.doi.org/10.4228/ZALF.DK.155 (accessed on 19 February 2021).

### 2.7. Statistical Analyses

#### 2.7.1. qPCR Data Analysis Via Self-ORGANIZING Map with Sammon Mapping

A combination of a SOM with SM was used to analyze the inoculation experiment’s qPCR results. SM had been introduced in 1969 by J.W. Sammon [62]. It aims to project n-dimensional data, with values for n > 2 observable on a 2D plane. The distances between any two data points in the 2D plane are as much as possible proportional to the dissimilarities in the n-dimensional data space. That would allow for visualization in 2D graphs with minimum loss of information. To that end, a projection on a 2D plane is iteratively optimized, starting from a random constellation by minimizing Sammon’s stress E (1):(1)$$E=\frac{1}{\sum_{i<j}\;d_{ij}^{*}}\;\sum_{i<j}\;\frac{(d_{ij}^{*}-d_{ij}{)}^{2}}{d_{ij}^{*}}$$
where $d_{ij}$ is the distance between two instances in the original data space, and $d_{ij}^{*}$ the distance between their projections. Different methods have been suggested for that optimization. However, they all share that there is no guarantee for convergence to the best possible solution. In particular, the algorithms reach their limits for large and high-dimensional data. Substantial improvements can be achieved by using upstream, powerful nonlinear projections methods. Especially SOM turned out to be exceptionally well suited in that regard.

SOM, also called Kohonen Feature Maps, have been introduced in the early 1980s [63,64]. Findings in neural science have stimulated the development of this type of an Artificial Neural Network. Correspondingly, SOM aim at mimicking certain aspects of visual information processing in human brains. Like SM, the purpose is an efficient low-dimensional projection of high-dimensional data sets, but following a different unsupervised learning approach.

SOM consists of units called neurons or codebook vectors arranged in a regular low-dimensional (often 2D) lattice. Each codebook vector consists of an n-dimensional vector where n is the dimension of the data set to be projected. Learning starts with the random initialization of the codebook vectors. Step by step, each instance of the original data set is compared with all codebook vectors. The most similar codebook vector (called winner neuron) is then slightly modified to adjust it a little more to the respective instance’s values. The so-called learning rate defines the degree of adjustment. The adjustment applies not only to the winner neuron but to the codebook vectors nearby, but the less, the more distant they are from the winner neuron. A neighborhood function defines the adjustment rate as a function of distance.

The same procedure is carried out for all instances of the data set and is repeated until a certain threshold is reached. Usually, learning is subdivided into two phases. During the first phase, the learning rate is relatively high, and the radius of the neighborhood function enables a rapid setup of a coarse structure. In the second phase of fine-tuning of the codebook vectors, both the learning rate and the neighborhood function’s radius are reduced to prevent overshooting. In the end, each instance of the data set can be assigned to a codebook vector with almost identical values. Besides, the SOM exhibits a smooth shape because adjacent codebook vectors are remarkably similar.

SOM are now increasingly used to classify large, high-dimensional data sets and often proved superior to conventional clustering approaches, e.g., for analysis of sequencing data [65,66,67]. Beyond that, Kohonen [64] suggested combining SOM and SM as a powerful approach of low-dimensional projection of large high-dimensional data sets where SM alone fails. Here the SOM output is used to initialize the SM algorithm that then performs an additional fine-tuning of the structure pre-defined by the SOM.

Compared to conventional 2D graphs, the data points’ location concerning the axes does not bear any information. Instead, distances between any two symbols in the graph are roughly proportional to dissimilarity in the original n-dimensional data space. Thus, SOM-SM can be compared to a sketch map that provides information about neighborhood relations rather than on absolute location. The same graph with the same spatial organization can illustrate different information provided by respective coloring or symbol types. Thus, a synopsis of various graphs with the same spatial structure serves as a compelling interface between large, high-dimensional data sets and the human brain, taking advantage of humans’ great capacity for visual pattern recognition.

It is the first time the method, known from neural network analysis, has been used to visualize large microbiological data. The SOM-SM analysis has been done using the R environment (R Core Team 2019), including the SOM [68] and MASS packages [69]. For the SOM, a rectangular grid with 11x8 neurons and a Gaussian neighborhood function was used. Learning comprised ten iteration steps in the first learning phase and 100 iteration steps in the second phase. For SM, 90 iteration steps were carried out.

#### 2.7.2. Mycotoxin Data Analysis Via Dunnett Multiple Comparison Procedure

The HPLC-MS/MS analysis data were tested for normal distribution (Kolmogorov– Smirnov test) and homogeneity of variances (Levene’s test). Differences between the different co-inoculation variants from the three sampling times and the control- inoculations were analyzed with a multiple comparison procedure according to Dunnett (α = 0.1, 0.05, and 0.01; [70]). Data for *Alternaria* mycotoxins ALT, AOH, AME, and TEN were not or at the limit detected and therefore not included in the analysis. Statistical tests were realized with OriginPro (Version 2019b; OriginLab Corporation, Northampton, MA, USA).

## 3. Results

The spray inoculation used has proven to be successful. All inoculated strains were detected in the corresponding samples with a high frequency and considerable amount of mycotoxins (for ears inoculated with fungal strains), which exceeded by far naturally occurring concentrations. In contrast, the control variants showed no or deficient presence. Random samples that were taken during the experiment confirmed that only the strains we used were detected in the ears. There was no contamination by other fusaria, alternaria, or pseudomonads. In almost all the samples, DON, 3-Ac-DON, 15-Ac-DON, DON-3G, NIV, ZEN, and TeA were recorded. For ALT, AOH, and AME, none or amounts around the LOD were present and therefore not included in the analysis. TEN was not detected in any of the samples. The control samples were not or slightly above the LOD of the analyzed mycotoxins during the whole experiment.

### 3.1. Abundance Analysis Via qPCR

#### 3.1.1. Comparison of the Average Abundances

##### *Alternaria At*625 Abundances

The co-cultivation of *At*625 followed by *Fg*23 (AF) or *Ps*9 (AP) did not affected the abundance of *Alternaria* within the first two weeks (Figure 3). After three weeks, however, a change in the dynamics was observed: *Fg*23 suppressed *At*625 by 60% (ACC: $1.8\times 10^{7}$ genome copy number (gcn) g${}^{-1}$dry matter (DM), AFF: $1.1\times 10^{7}$ gcn g${}^{-1}$DM). This effect becomes even more evident with a subsequent infection with *Ps*9, where the detected *At*625 gcn from $1.8\times 10^{7}$ g${}^{-1}$DM (ACC) decreased to $5.0\times 10^{6}$ gcn g${}^{-1}$DM (APP).

A previously infected wheat-ear with either *Fg*23 or *Ps*9 significantly reduced the chance of a subsequent infestation of *At*625. Two weeks after inoculation, *At*625 could not surpass the sprayed-on inoculation amount of about $2\times 10^{5}$ gcn g${}^{-1}$DM and seemed to be completely blocked in its establishment (CA-FA, CA-PA; Figure 3). After three weeks, *Fg*23 still stopped the *At*625 infection entirely (FAA; Figure 3). *Ps*9 as a primary colonizer also decreased the *At*625 establishment after three weeks significantly by 68% but not as efficiently as the *Fg*23 strain.

##### *Fusarium Fg*23 Abundances

The already established *Fg*23 strain was suppressed significantly by an upcoming *At*625 inoculation at the beginning (FC-FA) but recovered after three weeks (FCC-FAA). In the case of a previous infestation with *At*625 or *Ps*9, *Fg*23 was significantly suppressed in its establishment with an increased clarity after three weeks (CFF-AFF, CFF-PFF; Figure 3). *Ps*9 kept *Fusarium* back from growing right in the beginning by tenfold compared to the control (CF: $1.0\times 10^{6}$ gcn g${}^{-1}$DM, PF: $1.2\times 10^{5}$ gcn g${}^{-1}$DM). After week three, *Fusarium* was still unable to overcome the influence of *Pseudomonas* and reached only 30% of the control group (CFF-PFF; Figure 3).

##### *Pseudomonas Ps*9 Abundances

*Fg*23 promoted the growth of *Ps*9 when first applied (PC-PF) with a significant increase of almost 200%. However, after three weeks, this effect evens out and no differences to the control were detected (PCC-PFF; Figure 3). A follow-up inoculation with *At*625 showed the exact opposite pattern. After two weeks, no significant changes in the abundance of *Ps*9 could be detected (PC-PA), but after three weeks, *Ps*9 reacted to *At*625 with significant growth from $1.9\times 10^{8}$ gcn g${}^{-1}$DM (PCC) to $2.4\times 10^{8}$ gcn g${}^{-1}$DM (PAA).

An initial inoculation with *At*625 or *Fg*23 suppressed in both cases, *Ps*9 after two weeks significantly (CP-AP, CP-FP; Figure 3). This state did not change after three weeks, where the *Ps*9 abundance did not increase much (AP: $5.4\times 10^{6}$ gcn g${}^{-1}$DM, APP: $1.8\times 10^{7}$ gcn g${}^{-1}$DM; FP: $1.0\times 10^{7}$ gcn g${}^{-1}$DM, FPP: $1.8\times 10^{7}$ gcn g${}^{-1}$DM).

#### 3.1.2. Self-Organizing Map with Sammon Mapping

An effective way to visualize this high-dimensional data set context is a combination of SOM with SM that often supports superior to conventional clustering. It offers an efficient low-dimensional projection that takes advantage of humans’ visual pattern recognition capacity.

The three genera’s abundances over the experiment’s period are illustrated in a SOM-SM projection in Figure 4. Symbols indicate the single variants. Distance between any two symbols is inversely proportional to the similarity regarding *At*625, *Fg*23, and *Ps*9 abundance (r${}^{2}$ = 0.97).

Most symbols plot on one of the three axes, indicated by black arrows in Figure 4. Abundances increased over time, with the number of control periods. Correspondingly, variant A plots close to the center of the graph, variant AC more in the outward direction, and ACC close to the outer end of the respective arrow. The same applies to *Fg*23 (F, FC, FCC) and *Ps*9 (P, PC, PCC). On the other hand, variant C plots in the center of the graph, a subsequent inoculation shifts the variant more in an outward direction on one of the three axes (CA, CF, CP), and a second inoculation with the same genus reinforces that shift (CAA, CFF, CPP; Figure 4).

In any case, only genera that were intentionally inoculated established themselves. Thus, unintended contamination can be excluded. In addition, all replicates of the same variant plot close to each other (Figure 4, panels in lower two rows).

Most variants plot close to one of the three axes indicating that usually a second or third inoculation with another genus failed to establish. There were three exceptions and only after duplicate inoculation with a second genus (AFF, PAA, PFF; Figure 4). That holds for all replicates of the same variant.

### 3.2. Mycotoxin Analyses Via HPLC-MS/MS

#### 3.2.1. First Inoculation with *Fusarium Fg*23

An initial inoculation with *Fg*23 reduced the TeA production of the following *At*625 infection significantly within the first week (CA-FA). After three weeks, the effect reverses, and an even stronger TeA production is measured, compared to the control variant (CAA-FAA; Figure 5). *At*625 responded with an increase of TeA from 0.90 g g${}^{-1}$ (CAA) to 1.23 g g${}^{-1}$ (FAA) in the presence of *Fg*23.

*Fg*23 also reacted to the follow-up inoculation with *At*625. The most affected mycotoxin is ZEN, which is reduced by 90% both after two and three weeks (FC-FA and FCC-FAA; Figure 5). Furthermore, *At*625 noticeably reduced DON, 3-Ac-DON, 15-Ac-DON, and NIV in these inoculation-variants with an increasing effect after the third week (FCC-FAA). Subsequent infection with *Ps*9 hardly influenced the mycotoxin production of *Fg*23. Only ZEN showed a decline to 4% after three weeks (FCC-FPP; Figure 5).

#### 3.2.2. First Inoculation with *Alternaria*
*At*625

Samples that were first inoculated with *At*625 showed a significant reduction in the production of all measured *Fusarium* mycotoxins. After two weeks, they were generally at low levels (CF-AF). This effect was most significant for the DON derivatives, 3-Ac-DON (CF: 8.39, AF: 0.79) and 15-Ac-DON (CF: 8.12, AF: 0.66; Figure 5). After three weeks, *At*625 suppressed the entire mycotoxin production of *Fg*23 by significant 70% (CFF-AFF; Figure 5).

*Fg*23 on the other side, seemed to have no strong influence on the TeA of *At*625. Only in the variant AFF a distinct, but not significant, reduction of TeA was observed compared to the control variant ACC. The same response was observed for the variant APP, where *Ps*9 lowered the TeA production of *At*625 to 20% compared to the control (ACC-APP; Figure 5).

#### 3.2.3. First Inoculation with *Pseudomonas*
*Ps*9

A first inoculation with *Ps*9 caused a complete delay in the production of *Fg*23 mycotoxins in the first week after *Fg*23 was applied as the subsequent inoculant (CF-PF). After three weeks, *Ps*9 still affected DON, NIV, 3-Ac-DON, 15-Ac-DON, and DON-3G by more than halving the production in comparison to the control (CFF-PFF; Figure 5).

*Ps*9 also influenced *At*625 when applied first. After two weeks, *Ps*9 reduced the production of TeA from 0.15 g g${}^{-1}$ to 0.005 g g${}^{-1}$ and thus to 3% compared to the control(CA-PA). The effect is more evident after three weeks, when TeA decreased by about 90% compared to the control variant CAA (Figure 5).

Most *Fusarium* variants indicated lower mycotoxin production when inoculated second (XF, XFF with X = A or P; Figure 5). *Fg*23, on the contrary, could not restrain the TeA production of *At*625 over the three weeks. In contrast, *Ps*9, as an already established species, significantly reduced TeA production under 10% over the entire period (CA-PA, CAA-PAA; Figure 5). The only case in which the subsequent one also affected the mycotoxin production was an inoculation of *At*625 to an already *Fg*23 inoculated wheat-ear. After three weeks, *At*625 increased the TeA production and suppressed all *Fusarium* mycotoxins with significance in both DON derivatives and NIV (FCC-FAA; Figure 5). Therefore, *Alternaria* had the strongest influence on mycotoxin production during co-cultivation with *Fusarium*. Whereas, *Pseudomonas* showed its potential only when already established on the wheat-ear (PX, PXX with X = A or F; Figure 5).

### 3.3. Relationship between Fungal Growth and Mycotoxin Production during Competitive Interactions

*Fusarium* prevented *Alternaria* and *Pseudomonas* from establishing themselves and initially had a minor impact on *Alternaria’s* TeA production. However, *Alternaria* reduced the development of *Fusarium* when *At*625 was added to an existing *Fg*23 strain (FC-FA, FCC-FAA; Figure 3). *Alternaria* also had a negative effect on *Fusarium* mycotoxin production by significantly reducing the production of 3-Ac-DON, 15-Ac-DON, and NIV (FCC-FAA; Figure 5). Although *Alternaria* did not grow on *Fusarium*-covered wheat-ears, interactions still occurred at the mycotoxin level. *At*625 produced more TeA than in the control variants. Moreover, after three weeks, all *Fusarium* mycotoxins were reduced when *At*625 was inoculated first (CFF-AFF; Figure 5). *Pseudomonas* prevented neither *Alternaria* nor *Fusarium* from the subsequent establishment but still reduced the *Fusarium* and *Alternaria* abundances to some extent. Follow-up inoculation with *At*625 even increased the abundance of *Ps*9 (PCC-PAA; Figure 3) and significantly reduced the TeA production of *Alternaria* (CA-PA, CAA-PAA; Figure 5). As a consecutive inoculant, *Pseudomonas* also had a negative, though not significant, effect on the production of TeA (ACC-APP; Figure 5). However, both cases also reduced the abundance of *Alternaria* (CAA-PAA and ACC-APP; Fiugre Figure 3) compared to the control.

## 4. Discussion

Very little is known about the interaction mechanisms between fungi and bacteria during the infection process of the wheat-ears. Even less is known about the influence of the infection succession on plant community assembly. For this reason, this study focused on possible priority effects by the initial colonizer and the role of mycotoxins as biochemical weapons during this competitive process.

All three strains used showed different characteristics and strategies in dealing with consecutive co-inoculation. *Fg*23’s high competitive ability led to the suppression of *At*625 and *Ps*9, which failed to establish on wheat-ears that were already infected with *Fg*23. Mycotoxin production by *Fg*23 in the presence of its opponents remained fairly stable; however, it was unable to reduce the TeA production by *At*625. The priority effect of *At*625 showed the opposite behavior: this strain increased its mycotoxin production in the presence of *Fg*23 while it decreased that of its opponent, so long as it was initially there. As for growth, *At*625 suppressed both *Fg*23 and *Ps*9. However, only *Ps*9 was prevented from establishing if *At*625 was on the wheat-ear first. *Ps*9 showed an intermediate property in that it acted particularly well when it was first on the ear. Although not significantly, it affected the growth of the subsequent phytopathogenic fungi. Their mycotoxin production was also significantly reduced. The weakness of *Ps*9 was evident in the interaction against already established fungi, where it was unable to affect neither growth nor mycotoxin production. Moreover, it did not prevent the complete establishment of either of the two fungi. Thus, in terms of competitive strength, *Fg*23 ranked first, *At*625 second, and *Ps*9 third. The first colonizer affected the second colonizer in all combinations, whereas the expression depended on the strain’s fitness.

Study systems investigating interactions between *Alternaria* and *Fusarium* confirmed an effect on growth when co-inoculated. When *Fusarium* and *Alternaria* were grown on culture media in laboratory test systems, *Alternaria* was shown to suppress *Fusarium* growth [71,72]. In this particular study stystem, *Alternaria* spp. caused growth restriction of *F. graminearum* between 48–55% [72]. The opposite was found by Saß et al. [36] in liquid culture in an in vitro experiment: This showed that *F. graminearum* was able to suppress the growth of *A. alternata* significantly. In an in in vitro, fungal growth was also observed under consecutive co-cultivation [32]. It appears that the time advantage of the first inoculant allowed it to suppress the competitor [32]. This is consistent with the results we obtained in this study. Even though we did not find increased fungal growth during co-inoculation, the first colonizer reduced the growth rate of the subsequent one.

The situation was different for *Pseudomonas*, which increased its growth when subsequently inoculated with *Fusarium* or *Alternaria*. Our results contrast with those from the studies by Müller et al. [40,46]. In an antagonistic assay on natural substrates with selected antagonistic *Pseudomonas* isolates against *Alternaria* and *Fusarium* strains, fungal growth was slightly inhibited. After 13 days, no differences were observed to the controls [40]. Similarly, in the field trial where *P. simiae* was sprayed on ears of winter wheat, no effects of the bacterial antagonist on the growth of naturally occurring fusaria and alternaria were observed [46]. The response we observed could be due to the controlled inoculation of each strain and providing sufficient time for *Ps*9 to establish before another fungal strain was added.

Many studies often focus on antagonists and their influence on mycotoxin production by phytopathogens, in the context of a biocontrol approach for the natural suppression of mycotoxigenic pathogens in crop production [73,74,75]. Thus, studies focused mainly on *Fusarium*’s DON production during its interactions with antagonistic microbes [74,75,76,77,78]. Most reported that *F. graminearum* and *F. culmorum* reduced DON production in the presence of other microorganisms. The study by Gonzalez et al. [79] further demonstrated a general negative correlation between the presence of *A. alternata* and DON contamination in durum wheat. Müller et al. [32] also showed that a previous inoculation with *A. tenuissima* had a strong negative effect on DON production of a subsequent *Fusarium* inoculation. These results overlap with ours, in which the presence of *At*625, whether as a first or second inoculant, reduced DON production by *Fg*23.

The interactions between pseudomonads, fusaria, and alternaria, and their impact on mycotoxin production are also considered in several studies [40,46,77,80]. Experiments by Müller et al. [46] have shown that *P. simiae* interfered with the mycotoxin formation by fusaria and alternaria in vitro and a wheat field study. A reason for this could be that pseudomonads can produce secondary metabolites such as phenazine-1-carboxamide that are intensely antagonistic to fungal plant pathogens and suppress fungal growth, virulence, as well as mycotoxin biosynthesis [81]. A further in vitro experiment [40] revealed a distinctive reduction in DON, ZEN, and the *Alternaria* mycotoxins TeA, AOH, and AME up to complete ALT inhibition in the presence of *Pseudomonas* [40]. Although we did not detect AOH, AME, and ALT in our variants (most likely due to the *At*625 strain, which did not produce these mycotoxins even in the control variants), our results showed the same tendency. In the first inoculation with *Ps*9 and a second inoculation with *At*625, a significant reduction of TeA was detected over the whole period (PA/PAA; Figure 5). Furthermore, in a follow-up inoculation with *Fusarium*, DON and its derivatives were reduced after one week, as well as ZEN after two weeks (PF/PFF; Figure 5).

It is generally postulated that the co-cultivation of phytopathogenic *Fusarium* and *Alternaria* strains affects fungal growth and mycotoxin production. Contrary to other studies, in the results presented here, no long-term influence of *Fusarium* on mycotoxin formation by *Alternaria* was found. In the study by Müller et al. [32], a reduction of the *Alternaria* mycotoxins AOH, AME, and ALT were reported when co-cultivated with *F. culmorum* or *F. graminearum*. This correlation was attributed to the concurrent increased ZEN production, which was interpreted as an increased competitive response of *Fusarium* [32]. Especially ZEN, which is often counted among the antifungal traits, is mostly described to show a positive correlation in the presence of competitors [32,36,82].

Therefore, *Fusarium* strains seemed to dominate the *Alternaria* fungi, which might have been due to ZEN’s aid in the inter-fungal competition [33]. Our measurements of ZEN did not allow us to draw such a conclusion. In the presence of an opponent, whether inoculated before or after, there was always a decrease in ZEN production. However, it should be noted that ZEN production increased significantly after two weeks in the controls. It is possible that a different conclusion would have been reached if the experiment had been conducted over a longer period.

The plant’s involvement in the observed reactions is another aspect that should not be forgotten when considering the results. Several studies showed that plants could transform mycotoxins into their corresponding glucose or sulfate conjugates, referred to as masked mycotoxins. For example, the plant can transform DON to DON-3G [83] or DON sulfates [84]. Additional to this, it is assumed that the plant transports conjugated DON via membrane-bound transporters to vacuoles or apoplastic space [28]. Furthermore, glucose conjugates were also detected for ZEN and NIV in naturally infected samples [85,86]. Our study only analyzed DON-3G and found a ratio between DON and DON-3G of around 30%, as also described by other studies [87,88,89]. Since these results remained similar regardless of the variant, we assume that the reduction in mycotoxin with co-inoculation results from the interaction with the antagonists.

Interactions are key to a better understanding of community structure. For decades, ecologists aimed to understand how microbial communities are structured and how species promote that structure [90,91]. The underlying problem is that ecological communities are not static over time [92].

Priority effects, a term that in ecology was first introduced by Slatkin [93], describe biotic impacts that affect assembly history. This considers the extent to which organisms that arrive first at a habitat can influence the establishment, growth, or reproduction of species that arrive later [94,95,96].

In this study, we showed that the first inoculated species negatively influenced the subsequent ones. In all variants, the second colonizer’s growth was reduced, and the subsequent immigrant’s mycotoxin production, if affected, decreased. One exception to this was in the FAA variant, in which *Alternaria* was able to increase its TeA production due to a previous *Fg*23 inoculation.

Fukami [95] describes destabilizing mechanisms that lead to priority effects and divides them into two categories: niche preemption and niche modification. Niche preemption occurs in species with similar demands in both the requirement and the impact components of their niches. In this case, the first-arriving species reduces the available resources [95,97]. The later arriving species now achieves a much lower abundance with the limited resources they need for survival and reproduction. If niche preemption is very pronounced, an early arriving species may also prevent colonization by later-arriving species altogether [95,97]. Niche modification, on the other hand, is most pronounced between species with different requirements. The first species to arrive adds an impact component that leads to a modification of the local niches, making colonization by later-arriving species more difficult [95,98].

Priority effects that we observed may lead to the prevention of the co-existence of these species. However, because of these species’ evident co-occurrence in naturally infected wheat-ears [46,99,100], a more likely assumption is that the species are forced into a different niche, e.g., in different seed layers. We hypothesize on a niche preemption between the two fungi tested due to their strong growth in the wheat-ears after the artificial inoculation. Both *Fusarium* and *Alternaria* strains remarkably reduced the later arriving species’ growth. The high toxin production of fusaria or the even increased mycotoxin production of alternaria during the co-occurrence phase could also be considered to be a niche modification, preventing the other species from fully exploiting the niche. In this context, the consequences of the priority effect may be determined by enzymatic resource alteration, mycelial growth, and the formation of secondary metabolites such as mycotoxins [101]. Regarding *Ps*9, we also assume, based on our observations, that it can modify the niches for the fungi to inhibit their establishment. In the context of biocontrol, various antagonistic properties are described for pseudomonads: they can directly influence phytopathogens via antibiosis and competition for nutrients and, also indirectly, by stimulating defense mechanisms in the plants [73].

It is necessary to consider priority effects when applying a biological control agent against phytopathogenic fungi in agricultural practice [95]. In addition to the superficial pathogen-antagonist interactions, other microorganisms in the community must be taken into account. Interactions with non-target species, whether competitors, predators, or mutualistic partners, can increase the likelihood of priority effects through their species interactions [95,102]. Furthermore, it is equally important to understand the preceding dynamics in species composition and the timing of who is on the receiving end, and how newly arriving species compete.

## 5. Conclusions

The results of the present study implicated priority effects as important drivers in the infection process. Competitive interactions were found to be species-specific and differed in growth and mycotoxin production. The first colonizer generally retained an advantage, although the type of defense against newcomers differed. Our data suggest that there is a possibility to inhibit mycotoxin production by *Alternaria* and *Fusarium* through time-targeted applications of antagonistic pseudomonads. The perspective of microbial interactions influenced by priority effects provides a more holistic approach to biocontrol strategies. For this reason, future work will need to aim to identify more of these interactions.

## Figures and Tables

**Figure 1 microorganisms-09-00443-f001:**
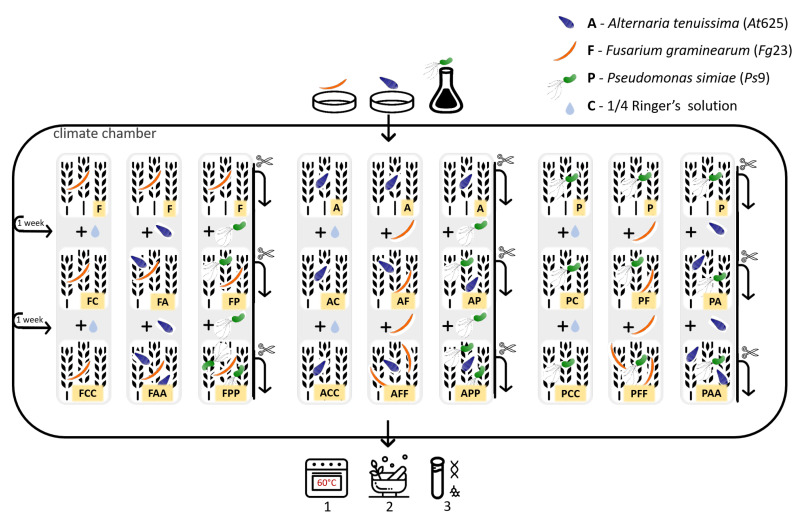
Experimental implementation - flow chart shows the inoculation process over three weeks. Corresponding symbols represent the inoculations with *At*625, *Fg*23 and *Ps*9. Suspensions were obtained from plates (for fungi) and liquid culture (for *Ps*9). As control inoculation, 1/4 sterile Ringer’s solution was applied. In the first week, only one isolate was applied to the wheat-ears. In the second week, an additional isolate different from the first one was applied. In the third week, the inoculation of the second week was repeated. Before each new inoculation, a part of the previous variant was harvested. All harvested wheat-ears were dried (1) and ground (2) before further analysis (3). The yellow highlighted box in the lower right corner shows the sample ID: with C = control with 1/4 sterile Ringer’s solution; A = *Alternaria tenuissima At*625, F = *Fusarium graminearum Fg*23, P = *Pseudomonas simiae Ps*9.

**Figure 2 microorganisms-09-00443-f002:**
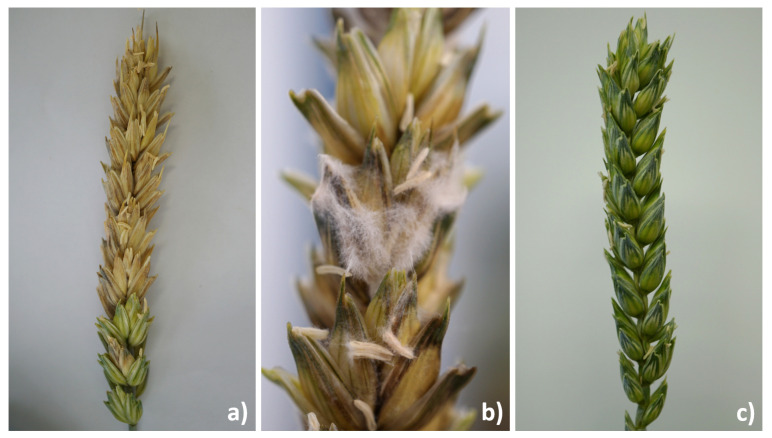
Wheat-ears symptom expression in a climate chamber experiment after 14 days of infestation: (**a**) distinct FHB disease symptoms on the ear; (**b**) fungal mycelium on the ears is visible; (**c**) symptom-free control wheat-ear (only treated with 1/4 strength Ringer’s solution).

**Figure 3 microorganisms-09-00443-f003:**
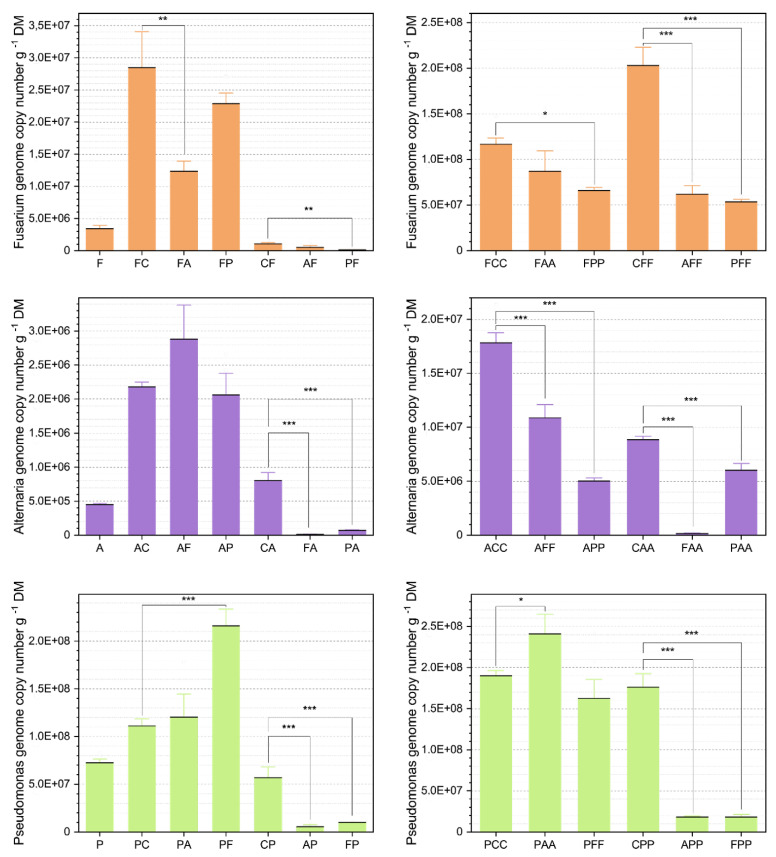
Fungal and bacterial growth were influenced by temporally shifted co-inoculation. Samples were taken after one (one capital letter), two (two capital letters), and three (three capital letters) weeks; with C = control with 1/4 sterile Ringer’s solution; A = *Alternaria tenuissima At*625, F = *Fusarium graminearum Fg*23, P = *Pseudomonas simiae Ps*9 inoculation. The mean values are plotted with their standard errors of the mean. Asterisk brackets indicate significant differences (* *p* < 0.1, ** *p* < 0.05, *** *p* < 0.01).

**Figure 4 microorganisms-09-00443-f004:**
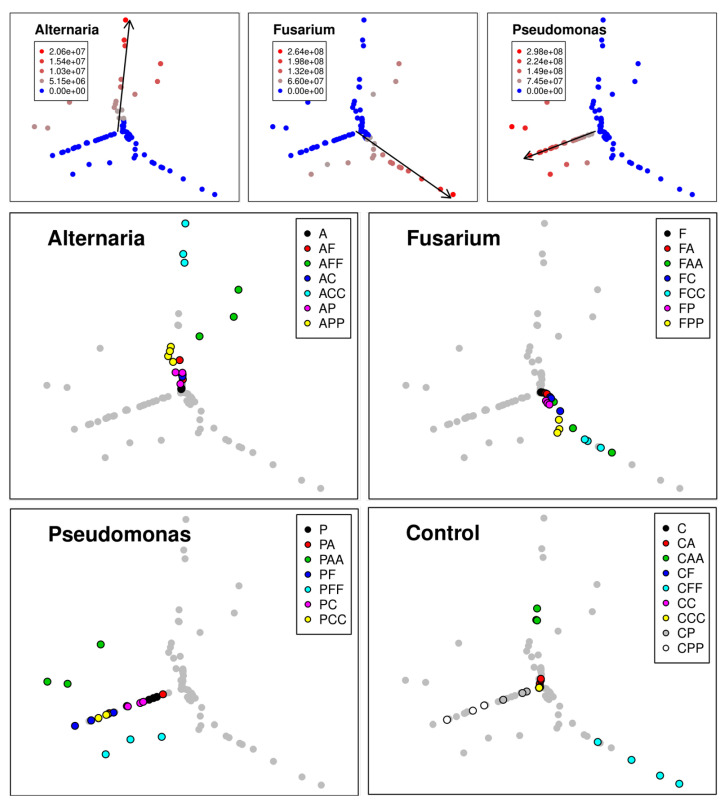
Abundance results of inoculation experiment illustrated in SOM-SM. Every single variant is indicated by a dot. Panels in the first row: Abundance of different genera. Black arrows indicate the direction of the steepest increase of abundance of the respective genus. Panels in lower rows: Different variants; C = control with 1/4 sterile Ringer’s solution; A = *Alternaria tenuissima At*625, F = *Fusarium graminearum Fg*23, P = *Pseudomonas simiae Ps*9 inoculation.

**Figure 5 microorganisms-09-00443-f005:**
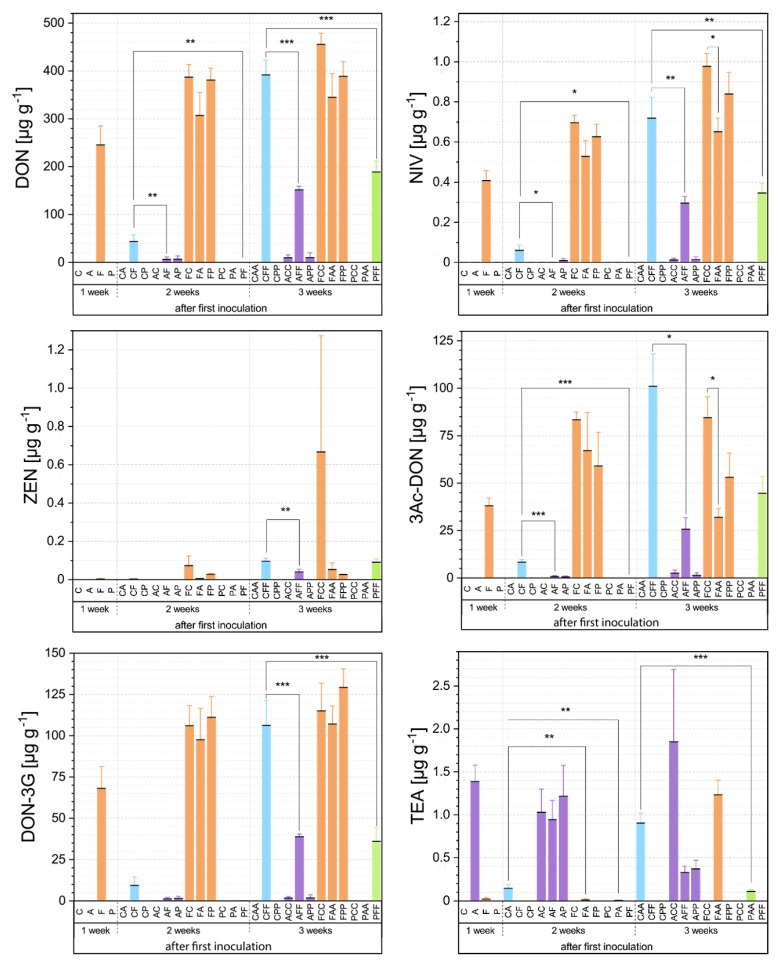
Bar plots of the mycotoxins of the inoculation experiment (DON = deoxynivalenol, NIV = nivalenol, ZEN = zearalenone, 3-Ac-DON = 3-acetyl deoxynivalenol, DON-3G = deoxynivalenol-3 glucoside, and TeA = tenuazonic acid). Chronical sequence of sampling is indicated by sample ID with C = control with 1/4 sterile Ringer’s solution; A = *Alternaria tenuissima At*625, F = *Fusarium graminearum Fg*23, P = *Pseudomonas simiae Ps*9. The mean values are plotted with their standard errors of the mean. Asterisk brackets indicate significant differences (* *p* < 0.1, ** *p* < 0.05, *** *p* < 0.01).

**Table 1 microorganisms-09-00443-t001:** Climate chamber program for the cultivation of summer wheat until flowering.

Time [Day]		0–6	7–15	16–24	25–48	49–59	60–66	67–End
day:	temperature [${}^{\circ }$C]	-	6	6	8	10	12	16
	humidity [%]	-	85	85	95	95	95	95
	duration of exposure [h]	-	12	12	14	14	14	14
	photosynthetic active	-						
	radiation [E/(m${}^{2}$s)]	-	276	326	415	413	413	420
night:	temperature [${}^{\circ }$C]	15	4	4	6	8	10	12
	humidity [%]	80	80	80	80	80	80	80
	duration [h]	24	12	12	10	10	10	10

**Table 2 microorganisms-09-00443-t002:** Primers used for the identification of *Pseudomonas*

Pse449	probe	5*’*- Fam-ACAGAATAAGCACCGGCTAAC-BHQ -3*’*
Pse435F	forward	5*’*- ACTTTAAGTTGGGAGGAAGGG -3*’*
Pse686R	reverse	5*’*- ACACAGGAAATTCCACCACCC -3*’*

**Table 3 microorganisms-09-00443-t003:** Optimized MS/MS conditions for detection of the target mycotoxins.

Analyte	Precursor Ion [*m/z*]	Product Ion [*m/z*]	DP ^*a*^ [V]	CE ^*b*^ [V]	CXP ^*c*^ [V]
NIV	371.1	59.1 ${}^{d}$	−45	−42	−7
	371.1	281.1	−45	−22	−15
${}^{13}C_{15}$-NIV	386.1	58.9 ${}^{d}$	−45	−42	−7
DON	355.1	59.2 ${}^{d}$	−40	−40	−8
	355.1	265.2	−40	−22	−13
${}^{13}C_{15}$-DON	370.1	279.1	−45	−24	−7
DON-3G	457.1	427.1 ${}^{d}$	−55	−16	−1
	457.1	247.1	−65	−25	−11
15-Ac-DON	397.1	337.1 ${}^{d}$	−40	−10	−9
	397.1	59.1	−40	−38	−8
3-Ac-DON	397.1	307.1 ${}^{d}$	−40	−20	−7
	397.1	59.1	−40	−38	−8
${}^{13}C_{17}$-3-Ac-DON	414.2	323.3 ${}^{d}$	−30	−24	−7
ZEN	317.1	131.1 ${}^{d}$	−80	−42	−8
	317.1	175.0	−80	−40	−18
${}^{13}C_{18}$-ZEN	335.2	140.2	−80	−34	−5
AOH	257.0	215.0 ${}^{d}$	−70	−36	−7
	257.0	147.0	−70	−37	−8
D3-AOH	260.1	216.0 ${}^{d}$	−80	−40	−8
	260.1	150.0	−80	−45	−8
ALT	291.0	203.0 ${}^{d}$	−75	−44	−17
	291.0	248.0	−75	−34	−7
D3-ALT	294.0	248.0 ${}^{d}$	−65	−35	−10
	294.0	203.0	−60	−40	−10
AME	271.0	256.0 ${}^{d}$	−40	−32	−13
	271.0	213.0	−40	−52	−16
D3-AME	274.0	256.0 ${}^{d}$	−40	−32	−13
	274.0	228.0	−40	−52	−16
TeA	196.2	139.1 ${}^{d}$	−70	−28	−7
	196.2	112.0	−70	−30	−9
TEN	413.2	141.0 ${}^{d}$	−50	−30	−7
	413.2	271.0	−50	−24	−7

^*a*^ Declustering potential; ^*b*^ Collision energy; ^*c*^ Cell exit potential; ^*d*^ Quantifier.

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
