# Peer review of "Co-Cultivation of Fusarium, Alternaria, and Pseudomonas on Wheat-Ears Affects Microbial Growth and Mycotoxin Production"

_microorganisms, 2021, doi:10.3390/microorganisms9020443_

Round 1

Reviewer 1 Report

This is a very well written manuscript with almost 0 mishaps. The novelty level is also quite high and shows great promise for the future especially considering the use of SOM+SM derived from neural network analysis. I also recognized undertakings in methodology where several simple but quite ingenious manoeuvres were used and that resulted in high quality data acquisition. Manuscript itself is stylized according to high standards and very scholastically. Again, some smaller mistakes are emphasized within the PDF file I uploaded and I encourage the authors to make amends where neccessary.

It was a pleasure and best of luck,

Anonymous barley breeder

P.S. From the point of view of application I would like to encourage the authors to try to forward their research with their colleagues from complementary fields of interest and collaborate in the future by shifting their focus to field conditions more when dealing with staple foods. In that way the scientific community would be able to see a more practical side of these findings. After all a measurable quantity, quality and the safety of food and feed is the true apex of scientific endeavours.

Author Response

Thank you for your revision. Please see the attached word file for our answers.

Reviewer 2 Report

This manuscript provided a piece of extensive information on “Co-cultivation of Fusarium, Alternaria, and Pseudomonas on Wheat-ears affects Microbial Growth and Mycotoxin Production. “  The information interesting. However, I do have a few questions and suggestions for revision before the manuscript gets accepted.

Below are some minor comments. Please note this is not a complete list. The authors should double-check their grammar and sentence construction. Some of the descriptions are not accurate or precise, which will need to be rephrased.

I would recommend professional English editing for this manuscript

Line 2: During….. Pseudomonas -  needs clarity

Line 5-6: Not clear

Line 19: The loss… possible – rewrite

Line 20-22: Plant protection plan? Explain

Line 25: FHB?

Line 31-33: Needs separate citation

Line 31-36: lacks recent citation

Line 82-85: More detailed… the wheat-ear – rewrite

Line 92-94: question or question (s)? - improve sentence construction

Line 107: This……Pathogens – rewrite

Discussion: needs to be more crisp

Figure 3: Needs more clarity regarding X in legends

Try to put more recent studies (citations) in Discission comparing the current results

I would recommend the publication of this manuscript after addressing minor changes.

Author Response

(The authors gave the same response as above.)
